# Efficient Recovery of *Candida auris* and Five Other Medically Important *Candida* Species from Blood Cultures Containing Clinically Relevant Concentrations of Antifungal Agents

Brunella Posteraro,[a,b] Giulia Menchinelli,[c] Vittorio Ivagnes,[a] Venere Cortazzo,[a] Flora Marzia Liotti,[c] Benedetta Falasca,[a] Barbara Fiori,[c] Tiziana D'Inzeo,[a,c] Teresa Spanu,[a,c] Giulia De Angelis,[a,c] Maurizio Sanguinetti[a,c]

aDipartimento di Scienze Biotecnologiche di Base, Cliniche Intensivologiche e Perioperatorie, Università Cattolica del Sacro Cuore, Roma, Italy
bDipartimento di Scienze Mediche e Chirurgiche, Fondazione Policlinico Universitario A. Gemelli IRCCS, Roma, Italy
cDipartimento di Scienze di Laboratorio e Infettivologiche, Fondazione Policlinico Universitario A. Gemelli IRCCS, Roma, Italy

Brunella Posteraro and Giulia Menchinelli, the first two authors, contributed equally to this article, and both should be considered first authors. Author order was determined randomly.
Giulia De Angelis and Maurizio Sanguinetti, the last two authors, contributed equally to this article, and both should be considered senior authors. Author order was determined randomly.

**ABSTRACT** *Candida auris* and other *Candida* species (*C. albicans*, *C. glabrata*, *C. parapsilosis*, *C. tropicalis*, and *C. krusei*) are important causes of bloodstream infection. Early or prolonged treatment with antifungal agents is often required. The inhibitory effect of antifungal agents in the patients' bloodstream may compromise the sensitivity of blood culture (BC) to diagnose and/or monitor patients with candidemia. Using a clinical BC simulation model, we compared antimicrobial drug-neutralizing BC media in BacT/Alert FA PLUS (FAP) or Bactec Plus Aerobic/F (PAF) bottles with non-neutralizing BC media in Bactec Mycosis IC/F (MICF) bottles to allow *Candida* growth in the presence of 100%, 50%, or 25% peak serum level (PSL) antifungal concentrations. In total, 117 organism/antifungal combinations were studied, and *Candida* growth was detected after incubating bottles into BacT/Alert VIRTUO or Bactec FX BC systems. Compared to control (without antifungal) bottles, both FAP and PAF bottles with 100% PSL antifungal concentrations allowed 100% recovery for *C. auris*, *C. glabrata*, and *C. parapsilosis*, whereas recovery was below 100% for *C. albicans*, *C. krusei*, and *C. tropicalis*. MICF bottles were less efficient at 100%, 50%, or 25% PSL antifungal concentrations, for all *Candida* species, except for *C. auris*. While azoles and amphotericin B did not hinder *Candida* growth in FAP or PAF bottles, echinocandins allowed *C. auris*, *C. glabrata*, and *C. parapsilosis* to grow in FAP, PAF, or MICF bottles. Overall, the maximum time to detection was 4.6 days. Taken together, our findings emphasize the reliability of BCs in patients undergoing antifungal treatment for candidemia.

**IMPORTANCE** While echinocandins remain the preferred antifungal therapy for candidemia, bloodstream infections caused by *C. auris*, *C. glabrata*, or, at a lesser extent, *C. parapsilosis* may be difficult to treat with these antifungal agents. This is in view of the high propensity of the above-mentioned species to develop antifungal resistance or tolerance during treatment. Azoles and amphotericin B are possible alternatives. Thus, optimizing the recovery of *Candida* from BCs is important to exclude the likelihood of negative BCs for *Candida* species, owing to the inhibitory effect of antifungal agents present in the blood sample with which BCs are inoculated. Consistently, our results about the recovery of medically important *Candida* species (including *C. auris*) from simulated BCs in BacT/Alert FAP, Bactec PAF, or Bactec MICF bottles containing clinically relevant antifungal concentrations add support to this research topic, as well as to the use of BCs for monitoring the clinical and therapeutic course of candidemia.

**KEYWORDS** antifungal agents, blood cultures, candidemia, *Candida* species, BacT/Alert bottles, Bactec bottles

Address correspondence to Maurizio Sanguinetti, maurizio.sanguinetti@unicatt.it.
The authors declare no conflict of interest.

**B**loodstream infections (BSIs) caused by *Candida* species, including the emerging *Candida auris* (1), are associated with high mortality rates (2), especially in the case of fungal resistance, i.e., when the invading fungus can grow in the presence of antifungal agents that would otherwise kill (fungicidal agents) or inhibit the growth of (fungistatic agents) the fungus *in vitro* (3). Unlike other pathogenic *Candida* species, such as *C. albicans*, *C. glabrata* (*Nakaseomyces glabrata*), *C. parapsilosis*, *C. tropicalis*, and *C. krusei* (*Pichia kudriavzevii*), which represent (in the order) the most frequent cause of BSI worldwide (2), *C. auris* has the uniqueness of exhibiting multidrug resistance (4, 5). Consistently, *C. auris* isolates (most from BSI cases) studied by Lockhart et al. (6) in 2017 showed resistance to 2 classes (azoles and polyenes) and, less frequently, to 3 classes (azoles, polyenes, and echinocandins) of antifungal agents. While a rare cause of candidemia in most hospitals in Europe and the United States exists, *C. auris* continues to spread across the world, thus posing a global health threat (7).

In the intensive care unit (ICU) setting, which provides the paradigm of *Candida* BSIs associated with poor outcome (8, 9), patients suspected of having candidemia or other forms of invasive *Candida* infection usually undergo a blood culture (BC) before starting antifungals, and then a reevaluation of antifungal therapy at day 5 (10). This is the time the patient's BC incubation in a BacT/Alert VIRTUO (bioMérieux) or Bactec FX (Becton, Dickinson) BC automated system to detect *Candida* growth is completed. If no *Candida* growth has been detected, antifungal therapy is usually stopped. However, a substantial number of critically ill patients (e.g., ICU patients with septic shock) are receiving early (e.g., before a BC is drawn) or prolonged (e.g., because of diagnostic uncertainties) treatment with antifungal agents (11). In both cases, optimizing the recovery of *Candida* from the patient's BC should be essential to increase the likelihood that a negative (or delayed positive) BC result is due to the inherent low sensitivity of BC systems (12), rather than the effect of antifungal agents present in the patient sample with which the BC is inoculated (13).

Here, we used spiked BCs with *C. auris*, or 5 other medically important *Candida* species to test antimicrobial drug-neutralizing BC media in BacT/Alert FA PLUS (FAP) or FN PLUS (FNP) bottles (bioMérieux) for the ability to allow *Candida* growth in the presence of clinically relevant concentrations of antifungal agents. The recovery rate from, and in the mean time to detection (TTD) with these media, were compared to those with neutralizing BC media in Bactec Plus Aerobic/F (PAF), Plus Anaerobic/F (PNF) bottles (Becton, Dickinson), or with non-neutralizing BC media in Bactec Mycosis IC/F (MICF) bottles (Becton, Dickinson), respectively.

## RESULTS

Our clinical *Candida* BC simulation approach in BacT/Alert or Bactec bottles used 6 *Candida* species (*C. auris*, *C. albicans*, *C. glabrata*, *C. krusei*, *C. parapsilosis*, and *C. tropicalis*), each in the presence of antifungal agents. Three concentrations (expressed as $\mu$g/mL) each of anidulafungin (7.2, 3.6, and 1.8), caspofungin (9.9, 5.0, and 2.5), micafungin (16.4, 8.2, and 4.1), fluconazole (14.0, 7.0, and 3.5; except for *C. auris* and *C. krusei*), posaconazole (3.3, 1.7, and 0.8), voriconazole (3.0, 1.5, and 0.8), or amphotericin B (3.5, 1.8, and 0.9; except for *C. auris*) were used to mimic, respectively, 100%, 50%, or 25% peak serum level (PSL) concentrations, which are achievable in patients who receive standard intravenous doses of antifungal agents (14). Therefore, 117 *Candida* organism/antifungal drug combinations (each tested in duplicate), corresponding to 234 simulated BCs (each replicated in 5 bottles), were studied (Table S1).

Initially, to mirror the routine clinical BC practice (15), we included 1,170 test (with antifungal) bottles, of which 468 (234 FAP and 234 FNP) were BacT/Alert bottles, and 702 (234 PAF, 234 PNF, and 234 MICF) were Bactec bottles. We also included 390 control (without antifungal) bottles, of which 156 (78 FAP and 78 FNP) were BacT/Alert bottles, and 234 (78 PAF, 78 PNF, and 78 MICF) were Bactec bottles. In keeping with *Candida* studies, which have *a priori* included aerobic (BacT/Alert or Bactec) BC bottles only (13, 16–19), no *Candida* growth was detected in 100% (312/312; 234 test and 78

control) BacT/Alert FNP bottles and in 97.4% (304/312; 234 test and 70 control) Bactec PNF bottles. The 8 *Candida* positive Bactec PNF (control) bottles grew *C. glabrata*, which is the only *Candida* species found to grow in anaerobic BC bottles until recently (20). After excluding 312 (234 test and 78 control) anaerobic bottles for each (BacT/Alert or Bactec) bottle type, our BacT/Alert versus Bactec bottles' comparison analysis included results for 312 BacT/Alert (FAP) bottles and 624 Bactec (PAF and MICF) bottles.

Table 1 and Fig. 1 show the overall recovery results for *Candida* species from BacT/Alert (FAP) or Bactec (PAF and MICF) bottles, respectively, according to the type (echinocandin, azole, or amphotericin B), or the PSL concentration of antifungal agents present in the bottles. As shown in Table 1, BacT/Alert FAP, Bactec PAF, and Bactec MICF control bottles allowed 100% recovery (78 bottles detected as positive/78 bottles tested) for all 6 *Candida* species included in the study. Regarding test bottles (Table 1), recovery was 100% in both BacT/Alert FAP (96/96) and Bactec PAF (30/30) bottles for *Candida* species grown in the presence of azole or amphotericin B antifungal agents. Conversely, recovery was 87.0% in BacT/Alert FAP (94/108), and 70.4% in Bactec PAF (76/108) bottles for *Candida* species grown in the presence of echinocandin antifungal agents. As expected, Bactec MICF bottles (which do not contain antimicrobial drug-neutralizing resins) allowed 50.0% (54/108), 72.9% (70/96), or 53.3% (16/30) recovery for *Candida* species grown in the presence of echinocandins, azoles, or amphotericin B, respectively. No recovery was noticed for *C. albicans*, *C. krusei*, or *C. tropicalis* in Bactec MICF bottles with echinocandins or for *C. glabrata* in Bactec MICF bottles with amphotericin B. As shown in Fig. 1, rates of *Candida* recovery in BacT/Alert FAP (68/78, 76/78, and 76/78), Bactec PAF (62/78, 68/78, and 72/78), and Bactec MICF (40/78, 50/78, and 50/78) bottles differed, depending on the antifungal concentrations (100% PSL versus 50% PSL or 25% PSL) present in the bottles. Using the McNemar's test, we found statistically significant differences between BacT/Alert FAP and Bactec PAF bottles ($P < 0.05$, for both 100% and 25% PLS comparisons; $P < 0.01$, for 50% PLS comparison), and between BacT/Alert FAP and Bactec MICF bottles ($P < 0.001$, for all PLS comparisons).

Tables 2, 3, and 4 show the recovery results for each *Candida* species from BacT/Alert (FAP) or Bactec (PAF and MICF) bottles according to the single antifungal agent (at the 100%, 50%, or 25% PLS concentration) present in the bottles. As shown in Table 2 (for echinocandins), BacT/Alert FAP and Bactec PAF bottles with anidulafungin did not allow growth for *C. albicans* (at all PSL concentrations), *C. krusei* (at the 100% PSL concentration), and *C. tropicalis* (at the 100% PSL concentration [FAP] or at all PSL concentrations [PAF]). Bactec PAF bottles with caspofungin did not allow growth for *C. albicans* (at both 100% and 50% PSL concentrations), *C. krusei* (at the 100% PSL concentration), and *C. tropicalis* (at both 100% and 50% PSL concentrations). BacT/Alert FAP bottles with micafungin (at the 100% PSL concentration) did not allow growth for *C. albicans* and *C. tropicalis*, whereas Bactec PAF bottles with micafungin did not allow growth for *C. albicans* (at the 100% PSL concentration) and *C. krusei* (at all PSL concentrations). In contrast, Bactec MICF bottles with anidulafungin, caspofungin, or micafungin (at all PSL concentrations) did not allow growth for *C. albicans*, *C. krusei*, and *C. tropicalis*. As shown in Table 3 (for azoles), Bactec MICF bottles did not allow growth for *C. albicans* with fluconazole or posaconazole (both at all PSL concentrations), *C. krusei* with posaconazole (at the 100% PSL concentration), *C. parapsilosis* with posaconazole (at all PSL concentrations), and *C. parapsilosis* with voriconazole (at all PSL concentrations). As shown in Table 4 (for amphotericin B), Bactec MICF bottles did not allow growth for all *Candida* species tested (*C. albicans*, *C. glabrata*, *C. krusei*, *C. parapsilosis*, and *C. tropicalis*) at the 100% PLS concentration, and for *C. glabrata* at both 50% and 25% PSL concentrations of antifungal agent. Fig. 2 shows the recovery rates in BacT/Alert FAP, Bactec PAF, or Bactec MICF bottles with antifungal (100% PSL, 50% PSL, or 25% PSL) concentrations for each of the 6 *Candida* species included in the study. Using the McNemar's test, we found statistically significant increases of recovery for *C. albicans* in Bactec PAF bottles ($P = 0.021$, for the 100% versus 25% PSL comparison), *C. krusei* in Bactec PAF or Bactec MICF bottles ($P = 0.045$ and $P = 0.014$, respectively, for all PSL comparisons), and *C. tropicalis* in BacT/Alert FAP bottles ($P = 0.045$, for all PSL comparisons).

**TABLE 1** Performances of BacT/Alert or Bactec blood culture bottles for the Candida species detection in the absence or presence of antifungal agents[a]

| Organism and condition used | BacT/Alert FAP bottles | | Bactec PAF bottles | | Bactec MICF bottles | |
|---|---|---|---|---|---|---|
| | Recovery, % (no. of bottles detected as positive/no. of bottles tested) | Mean time (h) to detection | Recovery, % (no. of bottles detected as positive/no. of bottles tested) | Mean time (h) to detection | Recovery, % (no. of bottles detected as positive/no. of bottles tested) | Mean time (h) to detection |
| **Candida species grown without antifungal agents** | | | | | | |
| C. auris | 100 (10/10) | 21.4 | 100 (10/10) | 21.2 | 100 (10/10) | 20.9 |
| C. albicans | 100 (14/14) | 21.5 | 100 (14/14) | 23.3 | 100 (14/14) | 19.9 |
| C. glabrata | 100 (14/14) | 17.8 | 100 (14/14) | 31.5 | 100 (14/14) | 17.5 |
| C. krusei | 100 (12/12) | 21.5 | 100 (12/12) | 27.1 | 100 (12/12) | 22.2 |
| C. parapsilosis | 100 (14/14) | 36.9 | 100 (14/14) | 31.5 | 100 (14/14) | 32.8 |
| C. tropicalis | 100 (14/14) | 20.5 | 100 (14/14) | 19.8 | 100 (14/14) | 19.3 |
| All species | 100 (78/78) | 23.4 | 100 (78/78) | 25.9 | 100 (78/78) | 22.1 |
| **Candida species grown with echinocandins** | | | | | | |
| C. auris | 100 (18/18) | 23.1 | 100 (18/18) | 34.1 | 100 (18/18) | 70.1 |
| C. albicans | 55.6 (10/18) | 40.0 | 33.3 (6/18) | 74.5 | 0 (0/18) | – |
| C. glabrata | 100 (18/18) | 18.0 | 100 (18/18) | 41.6 | 100 (18/18) | 20.5 |
| C. krusei | 88.9 (16/18) | 22.8 | 44.5 (8/18) | 42.6 | 0 (0/18) | – |
| C. parapsilosis | 100 (18/18) | 40.1 | 100 (18/18) | 34.8 | 100 (18/18) | 39.8 |
| C. tropicalis | 77.8 (14/18) | 26.8 | 44.5 (8/18) | 46.8 | 0 (0/18) | – |
| All species | 87.0 (94/108) | 27.7 | 70.4 (76/108) | 41.5 | 50.0 (54/108) | 43.5 |
| **Candida species grown with azoles** | | | | | | |
| C. auris | 100 (12/12) | 21.9 | 100 (12/12) | 22.7 | 100 (12/12) | 36.7 |
| C. albicans | 100 (18/18) | 22.1 | 100 (18/18) | 28.8 | 33.3 (6/18) | 97.8 |
| C. glabrata | 100 (18/18) | 19.1 | 100 (18/18) | 34.0 | 100 (18/18) | 20.4 |
| C. krusei | 100 (12/12) | 22.7 | 100 (12/12) | 29.8 | 83.3 (10/12) | 51.6 |
| C. parapsilosis | 100 (18/18) | 38.4 | 100 (18/18) | 33.1 | 33.3 (6/18) | 39.1 |
| C. tropicalis | 100 (18/18) | 21.9 | 100 (18/18) | 21.2 | 100 (18/18) | 41.9 |
| All species | 100 (96/96) | 24.6 | 100 (96/96) | 28.7 | 72.9 (70/96) | 41.4 |
| **Candida species grown with amphotericin B**[b] | | | | | | |
| C. albicans | 100 (6/6) | 22.9 | 100 (6/6) | 39.4 | 66.6 (4/6) | 51.0 |
| C. glabrata | 100 (6/6) | 18.6 | 100 (6/6) | 44.9 | 0 (0/6) | – |
| C. krusei | 100 (6/6) | 24.3 | 100 (6/6) | 31.7 | 66.6 (4/6) | 35.2 |
| C. parapsilosis | 100 (6/6) | 38.3 | 100 (6/6) | 42.9 | 66.6 (4/6) | 50.1 |
| C. tropicalis | 100 (6/6) | 24.5 | 100 (6/6) | 30.0 | 66.6 (4/6) | 49.7 |
| All species | 100 (30/30) | 25.7 | 100 (30/30) | 35.8 | 53.3 (16/30) | 46.5 |

[a]Growth of Candida species without or with antifungal agents (i.e., echinocandins, azoles, or amphotericin B) was detected after incubating the indicated blood culture bottles in the BacT/Alert VIRTUO or the Bactec FX blood culture automated system, respectively. In the case of recovery, the times to detection were recorded for each of Candida species allowed to grow in the absence or presence of antifungals agents. The symbol – indicates no recovery.
[b]The list of Candida species allowed to grow in the presence of amphotericin B does not include C. auris because of a MIC value of 4 $\mu$g/mL to the polyene antifungal drug (see Table S1) that led us to consider the organism resistant to amphotericin B, as already noted (3).

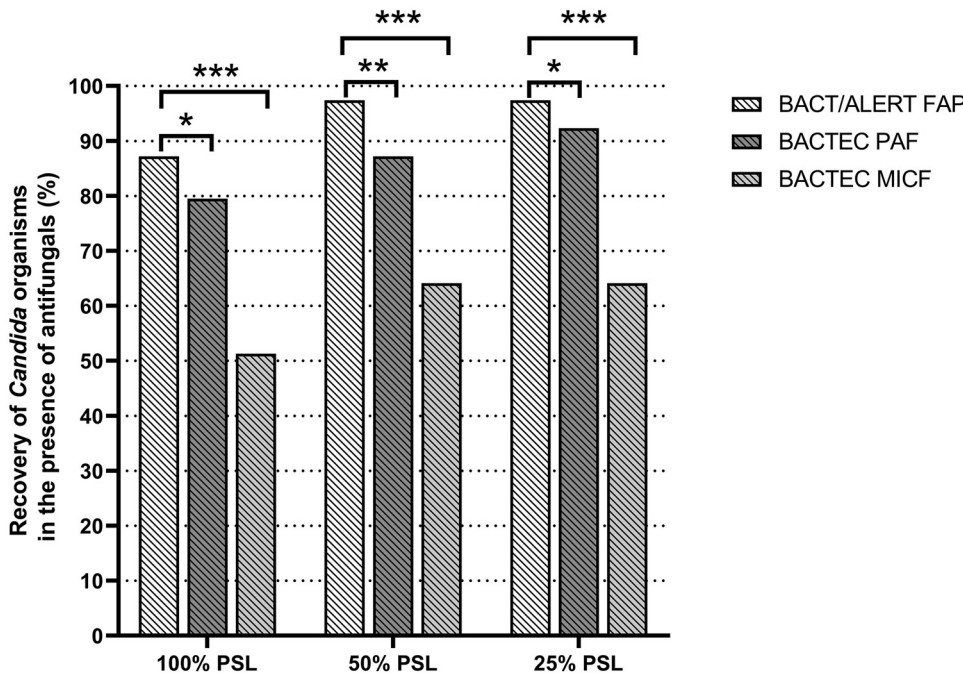

**FIG 1** Percentages of overall recovery for *Candida* species in BacT/Alert FAP, Bactec PAF, or Bactec MICF bottles by the antifungal concentrations present in each type of bottle. Comparisons between groups of bottles were assessed using the McNemar's test, which resulted in statistically significant differences (***, $P < 0.001$; **, $P < 0.01$; and *, $P < 0.05$). FAP, FA Plus; MICF, Mycosis IC/F; PAF, Plus Aerobic/F; PSL, peak serum level.

As also shown in Table 1, mean TTD (h) values for *Candida* species grown with echinocandins, azoles, or amphotericin B were, respectively, 27.7, 24.6, and 25.7 in BacT/Alert FAP bottles, 41.5, 28.7, and 35.8 in Bactec PAF bottles, and 43.5, 41.4, and 46.5 in Bactec MICF bottles. Based on ΔTTD (h) values (i.e., derived from the difference between the mean TTD values of test and control bottles for each antifungal class), these times were arbitrated to be slightly (ΔTTD, $< 2$ h), moderately (ΔTTD, 2 to 5 h), or highly (ΔTTD, $> 5$ h) delayed compared to those in bottles without antifungals ($P < 0.001$, for all comparisons; paired *t* test). Tables 2, 3, and 4 detail the mean TTD (h) values for each of 117 *Candida* organism/antifungal drug combinations in BacT/Alert FAP, Bactec PAF, or Bactec MICF bottles. For echinocandins (Table 2), the highest values noticed for *C. albicans*/micafungin were in BacT/Alert FAP or Bactec PAF bottles with the 50% PSL antifungal concentration (46.0 h and 89.6 h, respectively), or for *C. auris*/micafungin in Bactec MICF bottles with the 100% PSL antifungal concentration (110.5 h). For azoles (Table 3), the highest values (in bottles with the 100% PSL antifungal concentration) were noticed for *C. parapsilosis*/fluconazole in BacT/Alert FAP bottles (40.3 h), for *C. albicans*/posaconazole in Bactec PAF bottles (41.8 h), and for *C. albicans*/voriconazole in Bactec MICF bottles (108.4 h). For amphotericin B (Table 4), the highest values were noticed for *C. parapsilosis* in BacT/Alert FAP bottles with the 100% PSL antifungal concentration (39.8 h), for *C. glabrata* in Bactec PAF bottles with the 100% PSL antifungal concentration (49.4 h), and for *C. parapsilosis* in Bactec MICF bottles with the 50% PSL antifungal concentration (52.1 h).

## DISCUSSION

We showed that both BacT/Alert FAP and Bactec PAF bottles were highly (100%) efficient for detection of *C. auris*, *C. glabrata*, or *C. parapsilosis* in the presence of highest (100% PSL) concentrations of antifungal agents. The efficiency did not reach 100% with *C. albicans*, *C. krusei*, and *C. tropicalis*. In contrast, Bactec MICF bottles were considerably less efficient with all (excluding *C. auris*) the *Candida* species allowed to grow, and also with lowest (50% PSL or 25% PSL) concentrations of antifungal agents. Looking at the single antifungal agents used in the study, we noticed that azoles and amphotericin B did not

**TABLE 2** Recovery and time to detection of *Candida* species from BacT/Alert or Bactec blood culture bottles according to clinically relevant echinocandin antifungal concentrations[a]

| Antifungal drug and peak serum-level concn (%) used[b] | Organism used | BacT/Alert FAP bottles | | Bactec PAF bottles | | Bactec MICF bottles | |
|---|---|---|---|---|---|---|---|
| | | Recovery (no. of bottles detected as positive/no. of bottles tested) | Mean time (h) to detection | Recovery (no. of bottles detected as positive/no. of bottles tested) | Mean time (h) to detection | Recovery (no. of bottles detected as positive/no. of bottles tested) | Mean time (h) to detection |
| **Anidulafungin** | | | | | | | |
| 100 | *C. auris* | 2/2 | 24.3 | 2/2 | 47.2 | 2/2 | 79.9 |
| 50 | | 2/2 | 23.7 | 2/2 | 34.4 | 2/2 | 59.9 |
| 25 | | 2/2 | 21.3 | 2/2 | 29.7 | 2/2 | 43.9 |
| No drug | | 2/2 | 21.0 | 2/2 | 21.1 | 2/2 | 21.4 |
| 100 | *C. albicans* | 0/2 | – | 0/2 | – | 0/2 | – |
| 50 | | 0/2 | – | 0/2 | – | 0/2 | – |
| 25 | | 0/2 | – | 0/2 | – | 0/2 | – |
| No drug | | 2/2 | 21.2 | 2/2 | 23.3 | 2/2 | 20.3 |
| 100 | *C. glabrata* | 2/2 | 17.8 | 2/2 | 35.6 | 2/2 | 21.9 |
| 50 | | 2/2 | 17.6 | 2/2 | 33.1 | 2/2 | 20.7 |
| 25 | | 2/2 | 17.6 | 2/2 | 32.9 | 2/2 | 20.2 |
| No drug | | 2/2 | 17.8 | 2/2 | 32.1 | 2/2 | 18.1 |
| 100 | *C. krusei* | 0/2 | – | 0/2 | – | 0/2 | – |
| 50 | | 2/2 | 24.2 | 2/2 | 44.7 | 0/2 | – |
| 25 | | 2/2 | 22.2 | 2/2 | 44.1 | 0/2 | – |
| No drug | | 2/2 | 21.2 | 2/2 | 27.6 | 2/2 | 22.4 |
| 100 | *C. parapsilosis* | 2/2 | 40.2 | 2/2 | 33.4 | 2/2 | 35.0 |
| 50 | | 2/2 | 39.8 | 2/2 | 32.7 | 2/2 | 34.6 |
| 25 | | 2/2 | 39.2 | 2/2 | 33.8 | 2/2 | 34.7 |
| No drug | | 2/2 | 36.2 | 2/2 | 33.5 | 2/2 | 34.2 |
| 100 | *C. tropicalis* | 0/2 | – | 0/2 | – | 0/2 | – |
| 50 | | 2/2 | 29.7 | 2/2 | – | 0/2 | – |
| 25 | | 2/2 | 22.8 | 2/2 | – | 0/2 | – |
| No drug | | 2/2 | 20.8 | 2/2 | 18.6 | 2/2 | 18.8 |
| **Caspofungin** | | | | | | | |
| 100 | *C. auris* | 2/2 | 26.8 | 2/2 | 47.2 | 2/2 | 68.2 |
| 50 | | 2/2 | 22.9 | 2/2 | 34.4 | 2/2 | 63.3 |
| 25 | | 2/2 | 22.3 | 2/2 | 29.7 | 2/2 | 57.0 |
| No drug | | 2/2 | 21.2 | 2/2 | 21.1 | 2/2 | 20.4 |
| 100 | *C. albicans* | 2/2 | 39.1 | 0/2 | – | 0/2 | – |
| 50 | | 2/2 | 39.0 | 0/2 | – | 0/2 | – |
| 25 | | 2/2 | 37.5 | 2/2 | 56.8 | 0/2 | – |
| No drug | | 2/2 | 22.1 | 2/2 | 22.3 | 2/2 | 20.5 |
| 100 | *C. glabrata* | 2/2 | 18.5 | 2/2 | 41.6 | 2/2 | 22.0 |
| 50 | | 2/2 | 18.4 | 2/2 | 36.9 | 2/2 | 18.2 |
| 25 | | 2/2 | 18.2 | 2/2 | 35.8 | 2/2 | 18.3 |
| No drug | | 2/2 | 18.1 | 2/2 | 31.6 | 2/2 | 17.9 |
| 100 | *C. krusei* | 2/2 | 24.3 | 0/2 | – | 0/2 | – |
| 50 | | 2/2 | 19.8 | 2/2 | 50.5 | 0/2 | – |
| 25 | | 2/2 | 19.7 | 2/2 | 31.2 | 0/2 | – |
| No drug | | 2/2 | 20.5 | 2/2 | 27.4 | 2/2 | 21.9 |
| 100 | *C. parapsilosis* | 2/2 | 38.8 | 2/2 | 35.4 | 2/2 | 54.3 |
| 50 | | 2/2 | 38.8 | 2/2 | 34.5 | 2/2 | 45.0 |

**TABLE 2** (Continued)

| Antifungal drug and peak serum-level concn (%) used[b] | Organism used | BacT/Alert FAP bottles | | Bactec PAF bottles | | Bactec MICF bottles | |
|---|---|---|---|---|---|---|---|
| | | Recovery (no. of bottles detected as positive/no. of bottles tested) | Mean time (h) to detection | Recovery (no. of bottles detected as positive/no. of bottles tested) | Mean time (h) to detection | Recovery (no. of bottles detected as positive/no. of bottles tested) | Mean time (h) to detection |
| 25 | | 2/2 | 37.1 | 2/2 | 32.2 | 2/2 | 42.4 |
| No drug | | 2/2 | 36.5 | 2/2 | 32.1 | 2/2 | 33.1 |
| 100 | *C. tropicalis* | 2/2 | 27.7 | 0/2 | – | 0/2 | – |
| 50 | | 2/2 | 26.2 | 0/2 | – | 0/2 | – |
| 25 | | 2/2 | 22.1 | 2/2 | 38.3 | 0/2 | – |
| No drug | | 2/2 | 20.2 | 2/2 | 19.0 | 2/2 | 18.7 |
| 100 | Micafungin  *C. auris* | 2/2 | 22.9 | 2/2 | 35.9 | 2/2 | 110.5 |
| 50 | | 2/2 | 22.0 | 2/2 | 30.7 | 2/2 | 92.2 |
| 25 | | 2/2 | 21.5 | 2/2 | 27.8 | 2/2 | 55.8 |
| No drug | | 2/2 | 21.0 | 2/2 | 21.1 | 2/2 | 21.3 |
| 100 | *C. albicans* | 0/2 | – | 0/2 | – | 0/2 | – |
| 50 | | 2/2 | 46.0 | 2/2 | 89.6 | 0/2 | – |
| 25 | | 2/2 | 38.3 | 2/2 | 77.2 | 0/2 | – |
| No drug | | 2/2 | 21.8 | 2/2 | 24.5 | 2/2 | 19.5 |
| 100 | *C. glabrata* | 2/2 | 18.1 | 2/2 | 54.7 | 2/2 | 21.2 |
| 50 | | 2/2 | 17.9 | 2/2 | 54.3 | 2/2 | 21.2 |
| 25 | | 2/2 | 18.2 | 2/2 | 49.8 | 2/2 | 20.9 |
| No drug | | 2/2 | 17.9 | 2/2 | 31.1 | 2/2 | 17.7 |
| 100 | *C. krusei* | 2/2 | 24.3 | 0/2 | – | 0/2 | – |
| 50 | | 2/2 | 24.2 | 0/2 | – | 0/2 | – |
| 25 | | 2/2 | 24.1 | 0/2 | – | 0/2 | – |
| No drug | | 2/2 | 21.3 | 2/2 | 26.9 | 2/2 | 21.2 |
| 100 | *C. parapsilosis* | 2/2 | 42.9 | 2/2 | 38.8 | 2/2 | 40.5 |
| 50 | | 2/2 | 42.5 | 2/2 | 36.3 | 2/2 | 35.9 |
| 25 | | 2/2 | 42.0 | 2/2 | 36.2 | 2/2 | 35.8 |
| No drug | | 2/2 | 37.0 | 2/2 | 31.8 | 2/2 | 33.3 |
| 100 | *C. tropicalis* | 0/2 | – | 2/2 | 53.8 | 0/2 | – |
| 50 | | 2/2 | 29.9 | 2/2 | 48.3 | 0/2 | – |
| 25 | | 2/2 | 29.0 | 2/2 | 46.9 | 0/2 | – |
| No drug | | 2/2 | 20.8 | 2/2 | 19.5 | 2/2 | 20.7 |

[a]*Candida* species were allowed to grow in the presence of echinocandin antifungal agents in resin containing (BacT/Alert FAP and Bactec PAF) or non-containing (Bactec MICF) blood culture bottles, following bottles' incubation in the BacT/Alert VIRTUO or the Bactec FX blood culture automated system, respectively. In the case of recovery, the times to detection were recorded for each of the *Candida* species used in the study. The symbol – indicates no recovery.

[b]See text and Table S1 for details.

TABLE 3 Recovery and time to detection of *Candida* species from BacT/Alert or Bactec blood culture bottles according to clinically relevant azole antifungal concentrations[a]

| Antifungal drug and peak serum-level concn (%) used[b] | Organism used | BacT/Alert FAP bottles | | Bactec PAF bottles | | Bactec MICF bottles | |
|---|---|---|---|---|---|---|---|
| | | Recovery (no. of bottles detected as positive/no. of bottles tested) | Mean time (h) to detection | Recovery (no. of bottles detected as positive/no. of bottles tested) | Mean time (h) to detection | Recovery (no. of bottles detected as positive/no. of bottles tested) | Mean time (h) to detection |
| Fluconazole 100 | C. albicans | 2/2 | 23.4 | 2/2 | 28.2 | 0/2 | – |
| 50 | | 2/2 | 21.8 | 2/2 | 24.8 | 0/2 | – |
| 25 | | 2/2 | 21.1 | 2/2 | 24.7 | 0/2 | – |
| No drug | | 2/2 | 20.9 | 2/2 | 22.7 | 2/2 | 20.1 |
| 100 | C. glabrata | 2/2 | 19.7 | 2/2 | 36.7 | 2/2 | 21.4 |
| 50 | | 2/2 | 18.6 | 2/2 | 34.9 | 2/2 | 18.5 |
| 25 | | 2/2 | 19.0 | 2/2 | 34.0 | 2/2 | 18.1 |
| No drug | | 2/2 | 17.9 | 2/2 | 31.7 | 2/2 | 17.2 |
| 100 | C. parapsilosis | 2/2 | 40.3 | 2/2 | 32.3 | 2/2 | 50.8 |
| 50 | | 2/2 | 39.3 | 2/2 | 30.3 | 2/2 | 33.5 |
| 25 | | 2/2 | 38.2 | 2/2 | 30.1 | 2/2 | 33.2 |
| No drug | | 2/2 | 38.2 | 2/2 | 30.1 | 2/2 | 32.9 |
| 100 | C. tropicalis | 2/2 | 21.0 | 2/2 | 20.4 | 2/2 | 40.2 |
| 50 | | 2/2 | 20.3 | 2/2 | 20.7 | 2/2 | 29.4 |
| 25 | | 2/2 | 20.2 | 2/2 | 20.5 | 2/2 | 28.7 |
| No drug | | 2/2 | 19.7 | 2/2 | 20.1 | 2/2 | 18.2 |
| Posaconazole 100 | C. auris | 2/2 | 22.4 | 2/2 | 25.6 | 2/2 | 74.3 |
| 50 | | 2/2 | 22.8 | 2/2 | 24.7 | 2/2 | 50.0 |
| 25 | | 2/2 | 21.7 | 2/2 | 23.2 | 2/2 | 26.4 |
| No drug | | 2/2 | 22.4 | 2/2 | 22.1 | 2/2 | 21.1 |
| 100 | C. albicans | 2/2 | 23.4 | 2/2 | 41.8 | 0/2 | – |
| 50 | | 2/2 | 21.2 | 2/2 | 36.7 | 0/2 | – |
| 25 | | 2/2 | 21.0 | 2/2 | 28.2 | 0/2 | – |
| No drug | | 2/2 | 21.2 | 2/2 | 22.6 | 2/2 | 20.2 |
| 100 | C. glabrata | 2/2 | 20.0 | 2/2 | 33.0 | 2/2 | 21.4 |
| 50 | | 2/2 | 19.5 | 2/2 | 32.8 | 2/2 | 19.4 |
| 25 | | 2/2 | 19.4 | 2/2 | 31.6 | 2/2 | 19.2 |
| No drug | | 2/2 | 17.2 | 2/2 | 30.3 | 2/2 | 18.1 |
| 100 | C. krusei | 2/2 | 22.5 | 2/2 | 32.4 | 0/2 | – |
| 50 | | 2/2 | 22.2 | 2/2 | 32.1 | 2/2 | 69.1 |
| 25 | | 2/2 | 22.1 | 2/2 | 31.3 | 2/2 | 58.2 |
| No drug | | 2/2 | 21.8 | 2/2 | 28.5 | 2/2 | 22.9 |
| 100 | C. parapsilosis | 2/2 | 39.4 | 2/2 | 36.6 | 0/2 | – |
| 50 | | 2/2 | 38.3 | 2/2 | 36.2 | 0/2 | – |
| 25 | | 2/2 | 38.2 | 2/2 | 32.2 | 0/2 | – |
| No drug | | 2/2 | 37.9 | 2/2 | 30.7 | 2/2 | 31.8 |
| 100 | C. tropicalis | 2/2 | 23.2 | 2/2 | 24.6 | 2/2 | 54.8 |
| 50 | | 2/2 | 22.8 | 2/2 | 24.2 | 2/2 | 44.8 |
| 25 | | 2/2 | 22.5 | 2/2 | 24.1 | 2/2 | 39.5 |
| No drug | | 2/2 | 21.2 | 2/2 | 21.0 | 2/2 | 19.7 |

**TABLE 3** (Continued)

| Antifungal drug and peak serum-level concn (%) used[b] | Organism used | BacT/Alert FAP bottles | | Bactec PAF bottles | | Bactec MICF bottles | |
|---|---|---|---|---|---|---|---|
| | | Recovery (no. of bottles detected as positive/no. of bottles tested) | Mean time (h) to detection | Recovery (no. of bottles detected as positive/no. of bottles tested) | Mean time (h) to detection | Recovery (no. of bottles detected as positive/no. of bottles tested) | Mean time (h) to detection |
| Voriconazole | | | | | | | |
| 100 | C. auris | 2/2 | 21.4 | 2/2 | 21.1 | 2/2 | 26.8 |
| 50 | | 2/2 | 21.3 | 2/2 | 21.0 | 2/2 | 21.8 |
| 25 | | 2/2 | 21.2 | 2/2 | 20.7 | 2/2 | 21.0 |
| No drug | | 2/2 | 21.2 | 2/2 | 20.4 | 2/2 | 20.3 |
| 100 | C. albicans | 2/2 | 25.4 | 2/2 | 28.1 | 2/2 | 108.4 |
| 50 | | 2/2 | 20.9 | 2/2 | 23.8 | 2/2 | 99.2 |
| 25 | | 2/2 | 20.8 | 2/2 | 23.4 | 2/2 | 85.7 |
| No drug | | 2/2 | 20.6 | 2/2 | 23.4 | 2/2 | 20.1 |
| 100 | C. glabrata | 2/2 | 19.9 | 2/2 | 37.1 | 2/2 | 22.7 |
| 50 | | 2/2 | 18.0 | 2/2 | 33.3 | 2/2 | 21.9 |
| 25 | | 2/2 | 17.9 | 2/2 | 32.5 | 2/2 | 20.9 |
| No drug | | 2/2 | 17.9 | 2/2 | 31.7 | 2/2 | 17.2 |
| 100 | C. krusei | 2/2 | 24.3 | 2/2 | 28.6 | 2/2 | 61.8 |
| 50 | | 2/2 | 22.6 | 2/2 | 27.9 | 2/2 | 34.8 |
| 25 | | 2/2 | 22.5 | 2/2 | 26.6 | 2/2 | 34.3 |
| No drug | | 2/2 | 22.2 | 2/2 | 26.5 | 2/2 | 22.9 |
| 100 | C. parapsilosis | 2/2 | 37.5 | 2/2 | 37.3 | 0/2 | – |
| 50 | | 2/2 | 37.3 | 2/2 | 32.5 | 0/2 | – |
| 25 | | 2/2 | 37.3 | 2/2 | 30.2 | 0/2 | – |
| No drug | | 2/2 | 36.1 | 2/2 | 31.6 | 2/2 | 32.7 |
| 100 | C. tropicalis | 2/2 | 22.4 | 2/2 | 21.7 | 2/2 | 55.8 |
| 50 | | 2/2 | 22.3 | 2/2 | 21.7 | 2/2 | 42.8 |
| 25 | | 2/2 | 22.2 | 2/2 | 21.7 | 2/2 | 41.1 |
| No drug | | 2/2 | 19.6 | 2/2 | 19.9 | 2/2 | 19.2 |

[a]*Candida* species were allowed to grow in the presence of azole antifungal agents in resin containing (BacT/Alert FAP and Bactec PAF) or non-containing (Bactec MICF) blood culture bottles, following bottles' incubation in the BacT/Alert VIRTUO or the Bactec FX blood culture automated system, respectively. In the case of recovery, the times to detection were recorded for each of the *Candida* species used in the study. The symbol – indicates no recovery.

[b]See text and Table S1 for details.

**TABLE 4** Recovery and time to detection of Candida species from BacT/Alert or Bactec blood culture bottles according to clinically relevant amphotericin B antifungal concentrations[a]

| Antifungal drug and peak serum-level concn (%) used[b] | Organism used | BacT/Alert FAP bottles | | Bactec PAF bottles | | Bactec MICF bottles | |
|---|---|---|---|---|---|---|---|
| | | Recovery (no. of bottles detected as positive/no. of bottles tested) | Mean time (h) to detection | Recovery (no. of bottles detected as positive/no. of bottles tested) | Mean time (h) to detection | Recovery (no. of bottles detected as positive/no. of bottles tested) | Mean time (h) to detection |
| Amphotericin B 100 | C. albicans | 2/2 | 23.5 | 2/2 | 33.5 | 0/2 | – |
| 50 | | 2/2 | 22.7 | 2/2 | 27.5 | 2/2 | 51.9 |
| 25 | | 2/2 | 22.5 | 2/2 | 27.1 | 2/2 | 50.2 |
| No drug | | 2/2 | 22.2 | 2/2 | 24.6 | 2/2 | 18.7 |
| 100 | C. glabrata | 2/2 | 18.7 | 2/2 | 49.4 | 0/2 | – |
| 50 | | 2/2 | 18.7 | 2/2 | 43.2 | 0/2 | – |
| 25 | | 2/2 | 18.5 | 2/2 | 42.1 | 0/2 | – |
| No drug | | 2/2 | 17.4 | 2/2 | 32.1 | 2/2 | 16.2 |
| 100 | C. krusei | 2/2 | 25.7 | 2/2 | 35.4 | 0/2 | – |
| 50 | | 2/2 | 23.7 | 2/2 | 30.6 | 2/2 | 37.2 |
| 25 | | 2/2 | 23.6 | 2/2 | 29.1 | 2/2 | 33.2 |
| No drug | | 2/2 | 22.1 | 2/2 | 26.0 | 2/2 | 21.7 |
| 100 | C. parapsilosis | 2/2 | 39.8 | 2/2 | 46.9 | 0/2 | – |
| 50 | | 2/2 | 38.6 | 2/2 | 41.6 | 2/2 | 52.1 |
| 25 | | 2/2 | 36.5 | 2/2 | 40.3 | 2/2 | 48.2 |
| No drug | | 2/2 | 36.4 | 2/2 | 30.9 | 2/2 | 31.7 |
| 100 | C. tropicalis | 2/2 | 27.0 | 2/2 | 31.2 | 0/2 | – |
| 50 | | 2/2 | 24.3 | 2/2 | 30.6 | 2/2 | 50.7 |
| 25 | | 2/2 | 22.1 | 2/2 | 28.1 | 2/2 | 48.8 |
| No drug | | 2/2 | 21.0 | 2/2 | 20.4 | 2/2 | 19.4 |

[a]Candida species were allowed to grow in the presence of amphotericin B antifungal agent in resin containing (BacT/Alert FAP and Bactec PAF) or non-containing (Bactec MICF) blood culture bottles, following bottles' incubation in the BacT/Alert VIRTUO or the Bactec FX blood culture automated system, respectively. In the case of recovery, the times to detection were recorded for each of the Candida species used in the study. The symbol – indicates no recovery.

[b]See text and Table S1 for details.

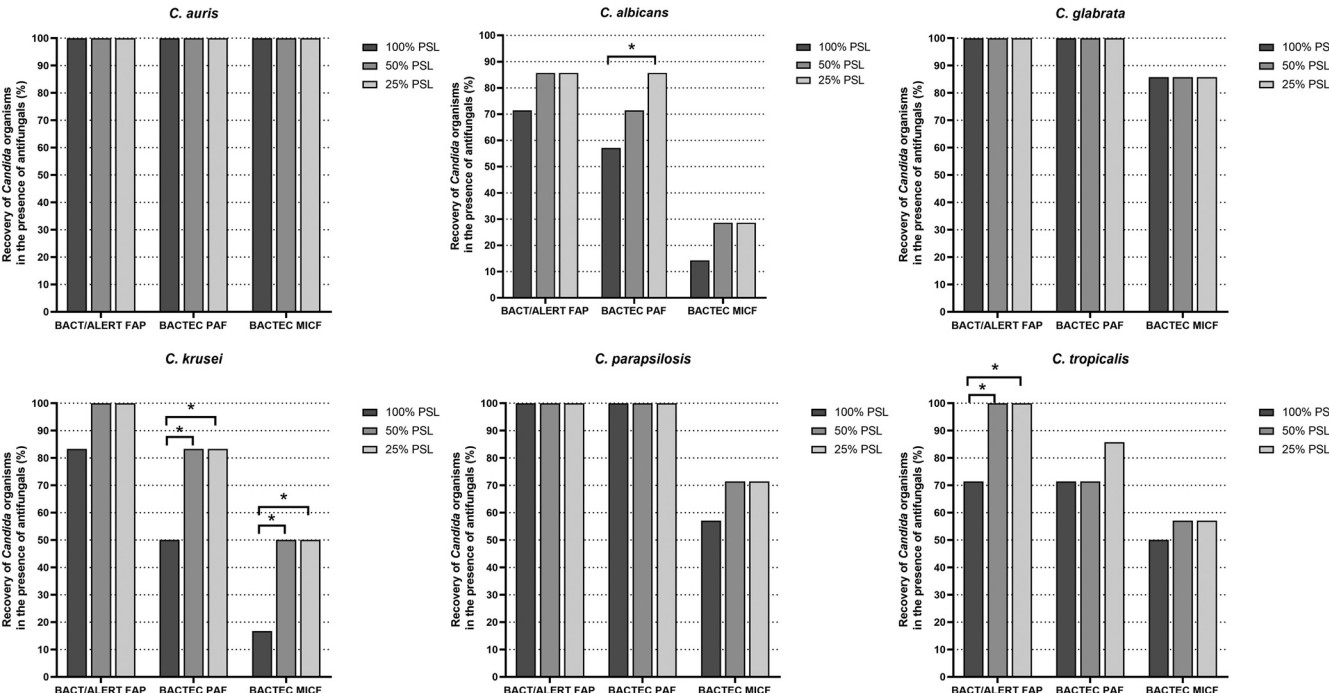

**FIG 2** Percentages of recovery for single *Candida* species in the presence of antifungal concentrations by BacT/Alert FAP, Bactec PAF, or Bactec MICF types of bottles. Comparisons between groups of bottles were assessed using the McNemar's test, which resulted in statistically significant differences (*, $P < 0.05$). FAP, FA Plus; MICF, Mycosis IC/F; PAF, Plus Aerobic/F; PSL, peak serum level.

hinder the growth of *Candida* species in both BacT/Alert FAP and Bactec PAF bottles. In contrast, we noticed that echinocandins did considerably affect the recovery of *Candida* species, which was apparent not only in Bactec MICF, but also in BacT/Alert FAP or Bactec PAF bottles. Overall, the maximum TTD (4.6 days) we recorded was nearly at the fixed upper limit (5 days) by the BacT/Alert or Bactec FX system's manufacturer.

Unlike the BacT/Alert FAP or Bactec PAF medium (15), Bactec MICF medium contains supplements that specifically adapt for the growth of *Candida* or other fungal species in BC bottles (18). In this context, it should be recalled that bioMérieux has adjusted the micronutrients in the BacT/Alert FAP medium to improve *Candida* detection from BC bottles. Consistent with our results, 2 studies (17, 18), independently, showed that fluconazole delayed the growth (17), whereas echinocandins or amphotericin B (but not fluconazole or voriconazole) hindered it (18) for *Candida* species (6 tested overall) in Bactec MICF bottles. The Köck et al. study (18) also revealed that echinocandins (particularly anidulafungin and micafungin) at the therapeutic peak serum concentration (i.e., the *Cmax*, which is nearly equivalent to the 100% PSL antifungal concentration in our study) did not allow reliable detection of *C. albicans* and *C. glabrata* in Bactec PAF bottles, to which Bactec MICF bottles were compared. In that study (18), recovery of both *Candida* species improved in Bactec (PAF and MICF) bottles with concentrations of 6.25% to 50% of the *Cmax* for echinocandins. Unfortunately, these values corresponded to the minimum serum concentrations (range, 1 to 6 $\mu$g/mL) achieved with the standard echinocandin therapy (18).

In both BacT/Alert FAP and Bactec PAF bottles, antimicrobial drug-neutralizing resins are likely to have allowed higher *Candida* recovery rates, in the presence of azoles or amphotericin B, than those in Bactec MICF bottles (Table 1). Therefore, Bactec MICF bottles may not be useful to detect candidemia or monitor *Candida* clearance from the bloodstream in patients undergoing treatment with these antifungal agents (21). However, *Candida* recovery rates were much less than 100% (albeit to a different extent) in all 3 types of BC bottles in the presence of echinocandins, suggesting that

the resins' ability to neutralize antifungal agents in BacT/Alert FAP and Bactec PAF bottles may not be sufficient with these antifungal agents. Of the 6 *Candida* species used in our study, *C. auris*, *C. glabrata*, and *C. parapsilosis* grew undisturbed in the presence of anidulafungin, caspofungin, or micafungin. The remaining 3 species (*C. albicans*, *C. krusei*, and *C. tropicalis*) did not grow at 100% PSL concentrations, or grew slowly (i.e., had a longer TTD) at 50% or 25% PSL concentrations of echinocandins (particularly anidulafungin), and this occurred more in Bactec MICF bottles than in Bactec PAF or BacT/Alert FAP bottles. Echinocandins are cyclic lipopeptides with hydrophobic fatty acid chains, which act as a "hook" for anchoring the drug in the fungal cell membrane, where it exerts an inhibitory action against the transmembrane enzyme beta-(1,3)-d-glucan synthase. Due to their chemical structure, echinocandins bind strongly (97 to 99%) to plasma proteins (22). In our experimental BC setting, protein-bound echinocandins could be presented as altered molecules for binding to polymeric resins—which are capable of binding to the hydrophobic regions of virtually any antimicrobial agent—and could, therefore, be less adsorbed by resins in the bottles compared to azoles or amphotericin B. Factors related to *Candida* growth kinetics (23) could have further influenced the interaction between polymeric resins and echinocandins, and this would have been evident for *Candida* species that grow faster or are morphologically more flexible.

Detection of *C. auris*, *C. glabrata*, and *C. parapsilosis* in BC bottles may be particularly important, considering the high propensity of these species to develop antifungal drug resistance or tolerance (the latter often termed "trailing growth" in clinical studies), following short-term antifungal exposure (3, 24), or in the presence of inhibitory or subinhibitory antifungal drug concentrations (25). Unlike azoles, echinocandins exert fungicidal activity against most *Candida* species but, remarkably, can paradoxically promote the growth *in vitro* of *C. parapsilosis* or other *Candida* species (such as *C. auris*) at concentrations above the MIC (7, 26). In our study, echinocandin antifungal effects, coupled with the absence (in Bactec MICF bottles) or the suboptimal (we suppose) ability (in BacT/Alert FAP or Bactec PAF bottles) of antifungal drug-neutralizing resins, may have prompted some *Candida* species to thrive in the presence of echinocandins. This occurred regardless of MICs for the species that were below the CLSI-established resistant breakpoints for echinocandin antifungals. Thus, echinocandins remain the preferred treatment for candidemia (21), particularly for *C. auris*, which is the only species with isolates shown to be pan-resistant to all 3 classes of antifungal drugs (1). Unlike previous studies (16, 18), the range of azole antifungal agents in our study included posaconazole, which may be a treatment chance in the case of *C. auris* BSI (27), as well as, like previous studies (16, 18), voriconazole, which is an alternative to fluconazole for the treatment of *C. glabrata* or *C. krusei* BSIs (21).

Our study has both strengths and limitations. To the best of our knowledge, this is the first study to use BacT/Alert FAP bottles to simulate BCs of patients receiving antifungal therapy for candidemia. However, we used only antifungal drug-susceptible organisms for each *Candida* species included in the study to appreciate the abilities of BacT/Alert FAP or Bactec PAF bottles' media to hinder or delay the fungistatic/fungicidal activity of antifungal agents in BC bottles. Additionally, our clinical BC simulation model was rigorous in terms of (i) volume of blood inoculated, (ii) antifungal drug concentrations mimicking the antifungal drug exposure level in patients, (iii) *Candida* inoculum, or (iv) incubation conditions in BacT/Alert or Bactec FX BC systems complying with the manufacturers' recommendations. However, we acknowledge that the median number of *Candida* organisms present in a *Candida* BSI episode may be ≤1 cell/mL (range, 0.1 and >1,000 cells/mL), a concentration of 0.5 to 1.0-fold below the yeast cell number tested by us. Thus, detection rates in BacT/Alert FAP, Bactec PAF, or Bactec MICF bottles spiked with *Candida* species might not mirror those in clinical BC bottles.

In conclusion, our study extends and confirms previous results about the recovery of medically important *Candida* species from simulated BCs in BacT/Alert FAP, Bactec

PAF, or Bactec MICF bottles containing clinically relevant concentrations of antifungal agents. While both BacT/Alert FAP and Bactec PAF bottles showed excellent performances with azoles and amphotericin B, 3 species (*C. auris*, *C. glabrata*, and *C. parapsilosis*) were recovered from all the BacT/Alert FAP, Bactec PAF, or Bactec MICF bottles with echinocandins. Our results emphasize the importance of surveillance BCs for the clinical and therapeutic monitoring of patients with candidemia.

## MATERIALS AND METHODS

**Yeast organisms and antifungal agents.** The yeast organisms included in the study consisted of 1 clinical isolate (*C. auris* Fondazione Policlinico Universitario A. Gemelli IRCCS [FPG]1), and 5 type/reference strains (*C. albicans* ATCC 90028, *C. glabrata* ATCC 2001, *C. krusei* ATCC 6258, *C. parapsilosis* ATCC 22019, and *C. tropicalis* ATCC 750) of *Candida* species. The *C. auris* FPG1 term means the "*C. auris* isolate one" at the FPG hospital of Rome (Italy), which is our study's location. Before testing, yeast organisms were retrieved from their frozen stocks, subcultured on Sabouraud dextrose agar (SDA) plates, and colonies were analyzed by matrix-assisted laser desorption/ionization time-of flight mass spectrometry (7, 28) to confirm the organism's identity to the species level. Antifungal agents (whose powders were provided by Toku-E) used in the study were as follows: anidulafungin (ANF), caspofungin (CSF), micafungin (MCF), fluconazole (FLZ), posaconazole (PSZ), voriconazole (VRZ), and amphotericin B (AMB). Prior to use in clinical BC simulation experiments (see below for details), each of the 6 *Candida* species was tested for susceptibility to ANF, CSF, MCF, FLZ, PSZ, VRZ, and AMB, which was performed according to the CLSI M27-A3 broth dilution reference method guidelines (29). Briefly, drug-free and yeast-free controls were included in 96-well microtiter plates, which were incubated at 35°C, and read visually after 24 h, whereas *C. krusei* ATCC 6258 and *C. parapsilosis* ATCC 22019 served as quality control strains, as recommended by the CLSI. The MIC endpoints were defined as the lowest antifungal drug concentration that caused a prominent decrease in or (only for AMB) a full inhibition of the visual growth relative to the drug-free control wells. As detailed in Table S1, for each *Candida* species, MICs were interpreted according to antifungal clinical breakpoints (CBPs) or, in the absence of CBPs, epidemiological cutoff values (ECVs), which the CLSI has established and reported, respectively, in the M60 (30) and M57S (31) documents. In the case of *C. auris*, for which CLSI CBPs/ECVs to azoles or AMB are lacking, azole MIC of $\geq$4 $\mu$g/mL (3) and AMB MIC of $\geq$2 $\mu$g/mL (27) were considered resistant, respectively. These MIC values were consistent with the tentative MIC breakpoints proposed by the CDC for *C. auris* and antifungal agents: FLZ, $\geq$32 $\mu$g/mL; AMB, $\geq$2 $\mu$g/mL; ANF, $\geq$4 $\mu$g/mL; CSF, $\geq$2 $\mu$g/mL; and MCF, $\geq$4 $\mu$g/mL (https://www.cdc.gov/fungal/candida-auris/c-auris-antifungal.html). It should be recalled that the modal MIC to FLZ among all *C. auris* isolates tested at the CDC was $\geq$256 $\mu$g/mL (as for our isolate; see Table S1). Nonetheless, the CDC proposed a FLZ MIC of $\geq$32 $\mu$g/mL, with the goal of capturing only those *C. auris* isolates that had a *Erg11* gene mutation-based associated azole resistance mechanism, and were, therefore, unlikely to respond to the FLZ antifungal agent. In parallel, ANF, CSF, MCF, FLZ, PSZ, VRZ, and AMB powders were dissolved in dimethyl sulfoxide (DMSO), as appropriate (28), to concentrations of 1,440 $\mu$g/mL, 1,980 $\mu$g/mL, 3,280 $\mu$g/mL, 2,800 $\mu$g/mL, 660 $\mu$g/mL, 600 $\mu$g/mL, or 700 $\mu$g/mL, respectively. The antifungal stock solutions were aliquoted and stored at $-$80°C prior to use in clinical BC simulation experiments, when a 1:10, 1:20, or 1:40 dilution series was prepared from each of 7 antifungal agents with 100%, 50%, or 25% PSL concentrations (14).

**Clinical *Candida* BC simulation model.** To make this model, we followed a previously developed protocol (19, 32) with some adaptations, as depicted in Fig. 3. After growth on SDA plates, fresh colonies of *Candida* (yeast) organisms grown on SDA were suspended in phosphate-buffered saline (PBS) to approximately 1 to 2 $\times$ 10$^6$ cells/mL (equivalent to a 0.5-McFarland standard density). The suspensions were diluted 10,000-fold in PBS, and aliquots (100 $\mu$L) of each suspension were plated on SDA to determine CFU counts after 48 h incubation of plates at 37°C. To simulate a candidemia level of 5 to 10 cells/mL, 0.5-mL *Candida* organism's suspension (containing 50 to 100 cells/mL) was used together with 9 mL human whole blood (obtained from the Transfusion Medicine Division of the FPG hospital), and 0.5 mL antifungal solution/PBS (see below) to fill each BC bottle with a final 10 mL injection volume. This volume corresponds to the optimal blood fill volume (8 to 10 mL, as per BacT/Alert or Bactec FX system manufacturer's instructions), to which the BC yield should be maximal (33). Unlike previous studies (13, 17) that used 1 to 5 (low inoculum) or 10 to 50 (high inoculum) *Candida* cells/mL in spiked BCs, we decided to use 1 inoculum, which represents a compromise between a lower and higher number of yeast cells that may be circulating in the blood during candidemia (33). As detailed in Fig. 3, simulated BC bottles were divided in 2 series, namely, test or control bottles, according to whether each antifungal drug concentration solution (0.5 mL; prepared as described above) was used in place of PBS (0.5 mL). We also included a negative (i.e., only blood-containing bottle) control for each simulated BC. In brief (Fig. 3), for any experimental condition (i.e., one of 117 *Candida* organism/antifungal drug combinations tested in total), 2 replicates of each bottle's type were separately filled using sterile precautions with each component (added in sequence) of the injection volume mentioned above (Fig. 3). All BacT/Alert or Bactec BC bottles were incubated, respectively, in a BacT/Alert VIRTUO BC system's (bioMérieux) or Bactec FX BC system's (Becton, Dickinson) instrument at 37°C for up to 5 days. The incubation period (h) for each bottle was recorded, and was used to calculate the TTD, i.e., the time elapsed from when the BC bottle was entered into the BC system instrument to when the bottle was flagged positive by the instrument. Additionally, for each *Candida* organism-antifungal combination tested, we calculated a $\Delta$TTD value,

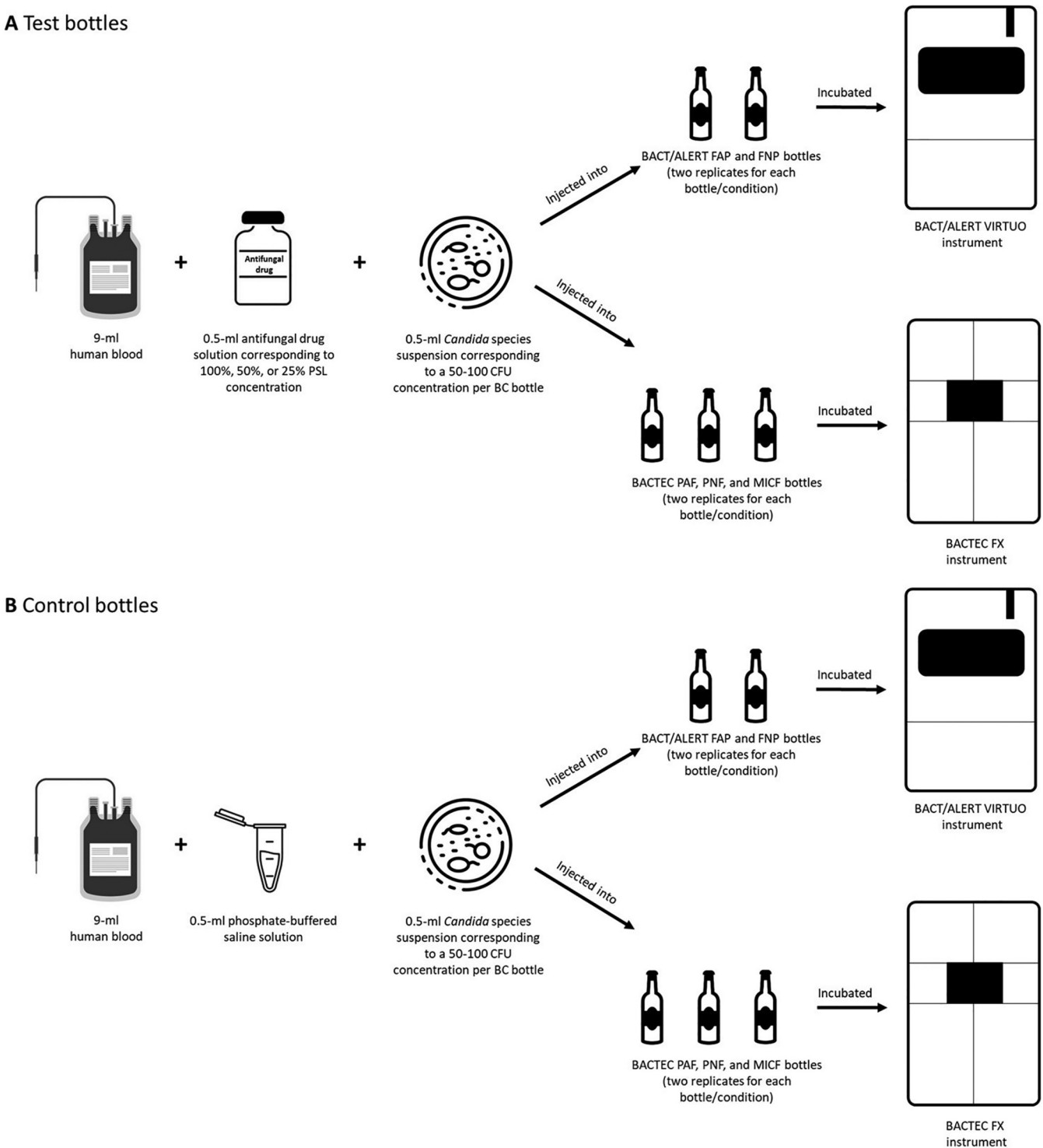

**FIG 3** Schematic diagram illustrating the protocol to obtain simulated *Candida* blood cultures with (A) or without (B) antifungal agents. Together with a *Candida* species, antifungal drug solution, which mimicked a given PSL concentration, or phosphate-buffered saline solution, was injected in a test (A) or control (B) blood culture bottle, respectively. For each experimental condition, replicates of test or control bottles were incubated, respectively, in the BacT/Alert VIRTUO or the Bactec FX blood culture system's instrument, until *Candida* growth was detected, or the manufacturer-recommended 5-day incubation period was completed. BC, blood culture; CFU, colony-forming unit; FAP, FA Plus; FNP, FN Plus; MICF, Mycosis IC/F; PAF, Plus Aerobic/F; PNF, Plus Anaerobic/F.

which was based on the difference between the test (with antifungal) bottle's mean TTD, and the control (without antifungal) bottle's mean TTD (Table 1). We arbitrarily categorized the mean TTD values of test bottles with echinocandins, azoles, or AMB as slightly ($\Delta$TTD, < 2 h), moderately ($\Delta$TTD, 2 to 5 h), or highly ($\Delta$TTD, > 5 h) delayed.

**Statistical analysis.** Results were reported as numbers and percentages for *Candida* organism recovery, or as a mean for TTD in the BacT/Alert BC or the Bactec FX BC systems, and differences between results in BC bottle groups were assessed using the McNemar's test or the paired *t* test, as appropriate. For all comparisons, the level of statistical significance was set at a *P* value of <0.05. The Intercooled Stata program version 11 and GraphPad Prism 7 were used to analyze data and/or to construct figures.

## SUPPLEMENTAL MATERIAL

Supplemental material is available online only.
**SUPPLEMENTAL FILE 1**, PDF file, 0.2 MB.

## ACKNOWLEDGMENTS

This work was presented, in part, at the 32nd European Congress of Clinical Microbiology and Infectious Diseases (ECCMID) held in Lisbon, Portugal (23 to 26 April 2022).

We are grateful to bioMérieux for providing reagents and funding this study, participating in the study design, and critically reviewing the manuscript before submission. In reviewing the manuscript, bioMérieux suggested only minor changes to the manuscript that had no impact on how the authors presented or interpreted the study results.

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
