## [Reviewer comments · Microbiology Spectrum]

Microbiology Spectrum

Efficient Recovery of *Candida auris* and Five Other Medically Important *Candida* Species from Blood Cultures Containing Clinically Relevant Concentrations of Antifungal Agents

Brunella Posteraro, Giulia Menchinelli, Vittorio Ivagnes, Venere Cortazzo, Flora Liotti, Benedetta Falasca, Barbara Fiori, Tiziana D'Inzeo, Teresa Spanu, Giulia De Angelis, and Maurizio Sanguinetti

Corresponding Author(s): Maurizio Sanguinetti, Fondazione Policlinico Universitario Agostino Gemelli IRCCS

Review Timeline:

Submission Date:	October 8, 2022
Editorial Decision:	December 4, 2022
Revision Received:	January 2, 2023
Accepted:	January 9, 2023

Editor: Alexandre Alanio

Reviewer(s): The reviewers have opted to remain anonymous.

Transaction Report:

DOI: <https://doi.org/10.1128/spectrum.04104-22>

December 4, 2022

Prof. Maurizio Sanguinetti
Fondazione Policlinico Universitario
Microbiology
L.go A. Gemelli 8
Rome, RM 168
Italy

Re: Spectrum04104-22 (Efficient Recovery of *Candida auris* and Five Other Medically Important *Candida* Species from Blood Cultures Containing Clinically Relevant Concentrations of Antifungal Agents)

Dear Prof. Maurizio Sanguinetti:

Both reviewers found the study well executed and useful for our community. They both pointed out that Biomerieux was involved in reviewing the manuscript. The authors should precise if the results were amended or modified after these reviewing or if it was only minor modifications. Please explain in details.

Link Not Available

Sincerely,

Alexandre Alanio

Journals Department
Reviewer comments:

Reviewer #2 (Comments for the Author):

The authors have investigated the efficiency of different blood culture media of the automated blood culture systems containing clinically relevant concentrations of antifungal drugs in isolation of six *Candida* species, including *Candida auris*. A clinical blood culture simulation model was used. Ultimately, the effect of the neutralizing resins on the existing antifungal drug was explored and differential data for different classes of antifungal drugs and varying *Candida* species were obtained. The presented data

are useful for daily medical practice, given the high number of patients receiving antifungal drugs and requiring blood culturing per their clinical manifestations. Some comments and inquiries:

1. Lines 112-116. The main conclusion of the study is that the neutralizing resins used may not be adequate for echinocandins but appear to neutralize the effect of azoles and amphotericin B in general. May the authors comment on this "further", considering the chemical structure of echinocandins?
2. Similarly, regarding the varying effects on different species of *Candida* - -What are the possible factors that may have influence on this? (growth rate, etc...) Adding an additional short discussion may be helpful.
3. Line 158. "Slightly", "moderately", or "highly delayed": Please insert exact definitions and relevant references at this point.
4. Lines 277-278. CDC tentative *C. auris* breakpoints should also be referenced and acknowledged at this point.

Reviewer #3 (Comments for the Author):

This is an interesting and well-executed study which provides valuable information on the reliability of blood culture in the presence of various antifungals. The authors are to be congratulated on the thoroughness of their methods.

I have a few small suggestions that I feel would be beneficial for the manuscript.

Candida glabrata and *Candida krusei* have relatively recently had name changes which are increasingly being used. Whilst I understand for familiarity the authors may choose to continue using their older names, I feel it would be helpful to also mention their newer names: *Nakaseomyces glabrata* and *Pichia kudriavzevii* respectively.

P5 - L87-91. It would make the reading of the manuscript slightly easier if the authors included the range of antifungal concentrations used as 100%, 50% and 25% PSL in the main body of the text as well as in the supplementary information.

Staff Comments:

Preparing Revision Guidelines

Please return the manuscript within 60 days; if you cannot complete the modification within this time period, please contact me. If you do not wish to modify the manuscript and prefer to submit it to another journal, please notify me of your decision immediately so that the manuscript may be formally withdrawn from consideration by Microbiology Spectrum.

Re: Spectrum04104-22 (Efficient Recovery of *Candida auris* and Five Other Medically Important *Candida* Species from Blood Cultures Containing Clinically Relevant Concentrations of Antifungal Agents)

Editor (Comment for the Author):

Both reviewers found the study well executed and useful for our community.

They both pointed out that bioMérieux was involved in reviewing the manuscript. The authors should precise if the results were amended or modified after these reviewing or if it was only minor modifications. Please explain in detail.

Answer: I have added a specification of how bioMérieux reviewed the manuscript via a short sentence in the Acknowledgments section. See page 15, lines 342 to 344 of the revised manuscript.

Reviewer #2 (Comments for the Author):

The authors have investigated the efficiency of different blood culture media of the automated blood culture systems containing clinically relevant concentrations of antifungal drugs in isolation of six *Candida* species, including *Candida auris*. A clinical blood culture simulation model was used. Ultimately, the effect of the neutralizing resins on the existing antifungal drug was explored and differential data for different classes of antifungal drugs and varying *Candida* species were obtained. The presented data are useful for daily medical practice, given the high number of patients receiving antifungal drugs and requiring blood culturing per their clinical manifestations. Some comments and inquiries:

Answer: While I am very grateful to the reviewer for appreciating the manuscript, I have tried to substantiate my responses to the comments/questions raised by the reviewer. This implied the addition of two references (nos. 22 and 23).

1. Lines 112-116. The main conclusion of the study is that the neutralizing resins used may not be adequate for echinocandins but appear to neutralize the effect of azoles and amphotericin B in general. May the authors comment on this "further", considering the chemical structure of echinocandins?

Answer: As suggested by the reviewer, I have added a few sentences to "further" comment on the main conclusion of the study, discussing how the chemical structure of the echinocandins may have affected the main findings of the study. See page 10, lines 213 to 220, with enclosed reference no. 22, of the revised manuscript.

2. Similarly, regarding the varying effects on different species of *Candida* - -What are the possible factors that may have influence on this? (Growth rate, etc...) Adding an additional short discussion may be helpful.

Answer: As in the previous comment, I elaborated on factors that may have affected the apparent species-specificity of the study findings. See page 10, lines 220 to 223, with enclosed reference no. 23, of the revised manuscript.

3. Line 158. "Slightly", "moderately", or "highly delayed": Please insert exact definitions and relevant references at this point.

Answer: As suggested by the reviewer, I have added details about how the categories were established by us. See page 8, lines 157 to 159, and page 14, lines 325 to 329 of the revised manuscript.

4. Lines 277-278. CDC tentative *C. auris* breakpoints should also be referenced and acknowledged at this point.

Answer: As suggested by the reviewer, I have added a few sentences (with enclosed reference) to acknowledge the CDC tentative *C. auris* breakpoints. See page 13, lines 287 to 294 of the revised manuscript.

Reviewer #3 (Comments for the Author):

This is an interesting and well-executed study which provides valuable information on the reliability of blood culture in the presence of various antifungals. The authors are to be congratulated on the thoroughness of their methods. I have a few small suggestions that I feel would be beneficial for the manuscript.

Answer: While I am very grateful to the reviewer for appreciating the manuscript, I have tried to substantiate my responses to the suggestions raised by the reviewer.

Candida glabrata and *Candida krusei* have relatively recently had name changes which are increasingly being used. Whilst I understand for familiarity the authors may choose to continue using their older names, I feel it would be helpful to also mention their newer names: *Nakaseomyces glabrata* and *Pichia kudriavzevii* respectively.

Answer: As suggested by the reviewer, I have mentioned the newer names of *Candida glabrata* and *Candida krusei*. See page 4, lines 55 to 56 of the revised manuscript.

P5 - L87-91. It would make the reading of the manuscript slightly easier if the authors included the range of antifungal concentrations used as 100%, 50% and 25% PSL in the main body of the text as well as in the supplementary information.

Answer: As suggested by the reviewer, I have added in the main text (as already reported in Table S1) the antifungal concentrations used as 100%, 50%, and 25% PSL, respectively. See page 5, lines 86 to 89 of the revised manuscript.

January 9, 2023

Prof. Maurizio Sanguinetti
Fondazione Policlinico Universitario Agostino Gemelli IRCCS
Microbiology
L.go A. Gemelli 8
Rome, RM 168
Italy

Re: Spectrum04104-22R1 (Efficient Recovery of *Candida auris* and Five Other Medically Important *Candida* Species from Blood Cultures Containing Clinically Relevant Concentrations of Antifungal Agents)

Dear Prof. Maurizio Sanguinetti:

Thank you for replying appropriately to reviewer's comments.

Your manuscript has been accepted, and I am forwarding it to the ASM Journals Department for publication. You will be notified when your proofs are ready to be viewed.

Sincerely,

Alexandre Alanio
Editor, Microbiology Spectrum
